# ArtSpeech: Adaptive Text-to-Speech Synthesis with Articulatory Representations

Submission Id: 2547

## ABSTRACT

We devise an articulatory representation-based text-to-speech (TTS) model, *ArtSpeech*, an explainable and effective network for human-like speech synthesis, by revisiting the sound production system. Current deep TTS models learn acoustic-text mapping in a fully parametric manner, ignoring the explicit physical significance of articulation movement. *ArtSpeech*, on the contrary, leverages articulatory representations to perform adaptive TTS, clearly describing the voice tone and speaking prosody of different speakers. Specifically, energy, $F_0$, and vocal tract variables are utilized to represent airflow forced by articulatory organs, the degree of tension in the vocal folds of the larynx, and the coordinated movements between different organs, respectively. We also design a multi-dimensional style mapping network to extract speaking styles from the articulatory representations, guided by which variation predictors could predict the final mel spectrogram output. To validate the effectiveness of our approach, we conducted comprehensive experiments and analyses using the widely recognized speech corpus, such as LJSpeech and LibriTTS datasets, yielding promising similarity enhancement between the generated results and the target speaker's voice and prosody. To promote reproducibility, we intend to make both the source code and the pre-trained model publicly available.

## CCS CONCEPTS

• **Human-centered computing**; • **Applied computing → Sound and music computing**;

## KEYWORDS

Text-to-Speech Synthesis, Style Transfer, Articulatory Representation

## 1 INTRODUCTION

Recently, text-to-speech (TTS) systems have made remarkable advancements [5, 35, 47] with deep learning techniques and NLP fundamental models [11, 19, 57]. However, due to their parametric nature, they suffer from several limitations: **First**, they lack explainability. As speaking style is implicitly modeled, the latent features are abstract and detached from the physical nature of speech production. **Second**, these TTS models typically extract style features on the frequency domain, paying less attention to modeling the

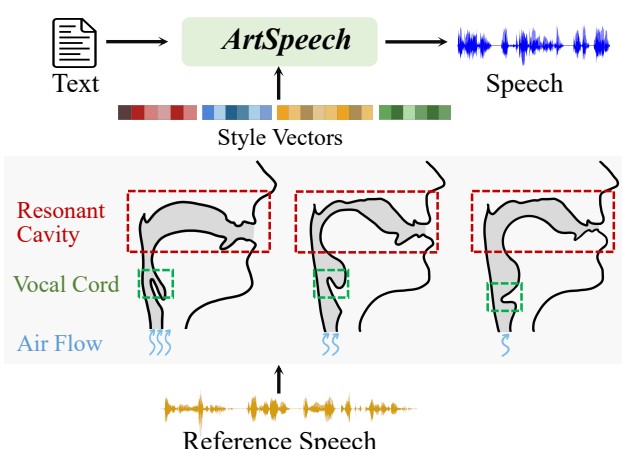

**Figure 1: Overview of our proposed *ArtSpeech*, an articulatory representation-based TTS synthesis model facilitating high voice tone and prosody similarity to the target speaker.**

articulatory system. Thus their controllability over style modification is limited. **Third**, each model employs a unique strategy to turn text into speech, requiring large-scale datasets to learn the parameters.

In light of the foregoing discussions, we are motivated to present *ArtSpeech*, an articulatory representation-based TTS synthesis network (Fig. 1). Human speech production, requiring the coordination of various organs, is a complex physiological process. The airflow from the lung vibrates through the vocal cord and then is shaped by the resonant cavity to produce sounds. With these organs' variations of shapes, sizes, and movement patterns, different speakers have their unique voices and prosody [15]. Drawing on the interpretability of speech production, our articulatory representation could serve as a generalizable and supplementary element for TTS frameworks; it is fully aware of the aforementioned limitations of parametric counterparts while showing better performance.

Several attempts have been made to explore leveraging articulatory data, *e.g.*, information recorded by magnetic resonant imaging (MRI) [44], ultrasound scan [13], and electromagnetic articulography (EMA) [54], to synthesize speech [1, 63] or improve the TTS process [39]. However, articulatory-based speech representation learning has been constrained by the complex modeling of physiological processes [3] and the quantity of available datasets [45], resulting in high word error rates of the synthesized speeches [63].

In this paper, we seek to investigate the following two research questions. First, what articulatory features are useful for TTS synthesis? To this end, we use three features that could be described the simplified human speech production: the intensity and velocity of the airflow expelled from the lungs, represented as *energy*; the

fundamental frequency at which the vocal cords oscillate, represented as $F_0$; articulatory movements of speech organs, represented as *vocal tract variables* (TVs). We design a TV extractor to achieve high-performance extraction and break the bottleneck of the limited dataset. We also propose a style mapping network to map articulatory features to speaking style.

The second research question we would like to explore is: How can speaking style be modeled using articulatory features, and to what extent can it enhance performance in the TTS task? We propose a multi-dimensional style mapping network to map articulatory features to the reference speeches' speaking style. These style vectors serve as guidance for the text-based variation predictors to predict both duration and articulatory parameters followed by the synthesis of mel spectrogram predictions.

Our extensive evaluation demonstrates that the generated speeches consistently outperform other TTS models in terms of voice tone and prosody similarity. For example, *ArtSpeech* demonstrates an improvement with a +0.05 of MOS on the single-speaker dataset (LJSpeech) and a +0.07 on the multi-speaker dataset (LibriTTS), compared to that of StyleTTS2 [36].

## 2 RELATED WORK

### 2.1 Articulatory-Based Speech Synthesis

As early as the 1790s, Kempelen *et al.* [27] developed a speaking machine, featuring a bellow to simulate the function of lungs, a flute to mimic vocal cords, and a tube serving as a stand-in for the mouth, thereby mimicking human speech production. After the 1970s, the articulatory system's physiological structure was further described through 2D geometry [38, 46] and 3D structural [14, 17, 61] modeling. These models enabled the simulation of the displacement and deformation of soft tissues while speaking yet resulting in limited quality and expressiveness of synthesized results [3].

With the most advanced medical equipment (*e.g.*, magnetic resonance imaging (MRI), ultrasound scan, and electromagnetic articulography (EMA)), researchers could capture real-time data on the movements of articulatory organs, leading to the emergence of data-driven TTS approaches [13, 44, 50, 54]. Specifically, GMM [55], HMM [39], and deep learning-based techniques [67] have been used to map articulatory features to waveforms [10, 63, 64] or as the intermediate representation to transfer linguistic features to acoustic features [41]. However, with these limited and low-quality articulatory datasets, the synthesized speeches are noisy and have a high word error rate [63]. *ArtSpeech*, on the contrary, aims to inherit the intuitive power of the articulatory system's physical significance, achieving high-performance speaking style modeling and breaking the bottleneck of limited articulatory data.

### 2.2 Expressive TTS

Among the numerous TTS algorithms, deep neural network-based methods are particularly remarkable, due to the improved expressiveness of synthesized speeches [47, 59]. Variance information (*e.g.*, duration, pitch and energy) are nonparametric and exemplar-driven and they are added to the phoneme hidden sequence, making results appealing and controllable [31, 33, 35, 47, 56]. Emotion

types [24, 34, 66] and textual styles [70] are a special form of variance information; it has been utilized to further refine the expression of synthesized speech. However, the consideration of variance information is limited, and intricate dynamics of speeches may not be well captured.

Global speaking style modeling of entire utterances is an alternative latent representation learning approach [7, 21, 52, 68]. A reference encoder, such as multi-head attention [53, 60] and variational autoencoders [29, 49], is always necessary to extract fine-grained latent vectors capturing speech prosody, style, and accent [48]. These methods, however, make controlling these style tokens a great challenge [21], not to mention the need for a large-scale training dataset [48]. In sharp contrast, *ArtSpeech* only uses articulatory representation for speaking style modeling and yields improved voice tone and prosody similarity between target speech and synthesized ones.

### 2.3 Adaptive TTS

In recent years, zero-shot speaker adaptation techniques have been proposed to mimic or preserve the unique characteristics of a speaker's voice. Many efforts initiated speaker representation learning, *e.g.*, AdaSpeech [6] and YourTTS [5], which incorporated speaker embeddings into TTS models (like VITS [28]) to facilitate multi-speaker TTS synthesis. On the other hand, large speech models also demonstrate their capabilities in such fields by leveraging diffusion models [51] and neural audio codecs [58, 69]. Overall, these approaches require extensive training data yet the limited articulatory data may present challenges.

As the great performance of AdaIN-based model [23] in the domain of image style transfer [8, 26], *i.e.* leveraging style vectors to disentangle the feature space, there has been a recent surge of interest in considering a similar strategy in adaptive TTS. For example, Meta-StyleSpeech [42] and StyleTTS [35] explored to extract implicit style vectors from the target mel spectrogram. Several approaches such as StyleTTS2 [36] and GenerSpeech [22] have made an in-depth study of the style vector space to achieve more controllable speech synthesis results. Therefore, we desired to explore using an AdaIN-based model to decouple multiple style vectors from target speakers' speeches – no extra articulatory data is needed, model controllability is enhanced, and the similarity between synthesized results and the target speech is improved.

## 3 ARTSPEECH

This section introduces the formulation of our *ArtSpeech* comprising articulatory representation learning, multi-dimensional style extraction, and multiple parameter predictors. The training process is also introduced here.

### 3.1 Overall Workflow

The goal of the standard TTS model is to accurately convert text content *c* into natural-sounding speech *s*. In adaptive TTS, the reference speech *r* is defined as the speaking style template (including both voice tone and prosody) that the synthesized speech should express. As human speech production can be systematically decomposed into three foundational stages, *i.e.* respiration, phonation, and articulation [9], we represent them with three articulatory features

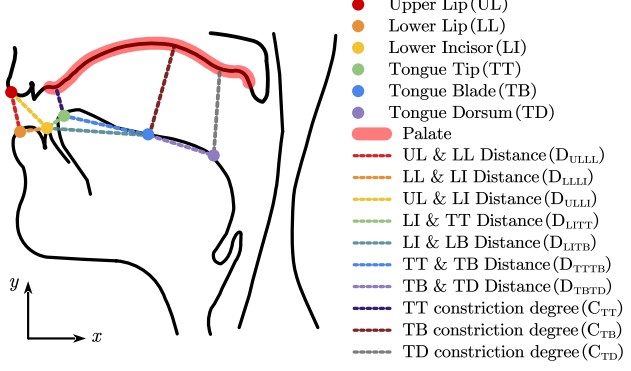

**Figure 2: EMA data (points) and vocal tract variables (dashed lines) calculated using relative distances in EMA.**

– energy $e$, fundamental frequency (F$_0$) $f_0$, and vocal tract variables (TVs) $v$. Our style extractor $SE$ first performs articulatory feature extraction from $r$, which along with $r$ are further mapped to style vectors $sv$.

The style vectors $sv$ are then incorporated into the TTS model as additional input alongside the $c$. This can be formulated as follows:

$$se = SE(e, f_0, v, r), s' = TTS(c, sv),$$

where $s'$ represents the spoken output of $c$ with a speaking style similar to $r$.

### 3.2 Articulatory Representation

We use three articulatory features to model the human speech production process.

• First, the lungs expel air, providing power for pronunciation. The volume and velocity of expelled air are typically proportional to the energy of the sound [43]. Thus, we choose energy as an articulatory feature to describe the respiration stage.

• Second, the airflow impacts the vocal cords, causing them to vibrate. The frequency of these vibrations directly determines the fundamental frequency (F$_0$)[1] of the speech [4].

• Finally, the resonant cavities formed by articulatory organs (from the larynx to the lips) modulate formant frequencies [12, 15], *i.e.* the movement of organs dynamically sculpt vocal tract shapes, enabling the production of a diverse set of phonemes and a wide variety of voice tones.

The relationship between vocal tract shape and formant frequencies can be represented by considering factors such as tongue position, lip shape, jaw aperture, vocal tract length, and the areas of both anterior and posterior cavities of the vocal tract [15, 32]. The transfer function method [15] and impedance phase shift method [37] have been proposed to simulate the relationship between vocal tract area function and the vowel formant frequency, but causing a complex solving process and are not easily applied to more intricate syllables. Wu *et al.* [62], conversely, proposed using EMA-recorded data for speech inversion and showed that the real-time relative

---

[1]A similar concept is 'Pitch', which describes how our ears and brains interpret the signal, yet F$_0$ describes the actual physical phenomenon.

distances between vocal organs could effectively enhance performance. Inspired by this, we incorporate such real-time relative distances into our speaking style modeling.

As shown in Fig. 2, we leverage the real-time recorded six articulatory points, *e.g.*, the tongue tip (TT), tongue blade (TB), etc., by electromagnetic articulography (EMA) technique and define 10 distances and constriction degrees accordingly [16]. For example, the movement of the tongue tip could be indicated by $D_{\text{LITT}}$ and $D_{\text{TTTB}}$, where $D_{\text{TTTB}}$ is defined as

$$D_{TTTB}[t] = \|TT[t] - TB[t]\|_2,$$

where $t$ is the timestamp. Similar definitions apply to $D_{ULLL}$, $D_{ULLI}$, and $D_{LLLI}$, indicating lip convexity and opening; $D_{LITB}$ and $D_{TBTD}$, indicating tongue position and vocal tract length respectively. $C_{TT}$, $C_{TB}$, and $C_{TD}$ represent constriction degrees, indicating the shortest distance between the 3 tongue positions and the palate curve, respectively. Take $C_{TT}$ as an example, *i.e.*

$$C_{TT}[t] = \min_{x,y}\|TT[t] - pal(x,y)\|_2,$$

Where $pal(\cdot)$ represents the point on the palate curve, with $x$ and $y$ denoting the plane coordinate system formed by the sagittal plane of the vocal organs. Note that the palate location is pre-recorded as it is relatively static to the head pose while speaking [54].

### 3.3 Style Extraction

Similar to Fastspeech2 [47], we leverage the norm of the mel spectrogram to approximate the energy of the speech; the JDC network [30] to estimate the F$_0$ sequence from the mel spectrogram.

Then, we propose a *TV extractor*, as shown in Fig. 3(a), to estimate the 10-d vocal track variables from the mel spectrogram of $r$ along with the extracted energy and F$_0$. The extractor consists of five conformer blocks [18] and one bi-LSTM [20] layer, allowing for the integration of both global and local information and capturing short-term correlation of vocal track changes. We pre-trained *TV extractor* with L1 loss on HPRC dataset [54], which has 7.9 hours of 44.1 kHz speeches and 100 Hz EMA data recorded by 8 participants.

With the foregoing extracted phoneme-level articulatory features, we further design a multi-dimensional *Style Extractor* to model the speaking style of the reference speech. As shown in Fig. 3(b), five separate mapping networks are designed to obtain style vectors accordingly, *i.e.* energy, F$_0$, TVs, duration, and mel spectrogram. These mapping networks have similar structures, consisting of multiple residual blocks, and take the sentence-level feature vectors as input and output a fixed-length style vector. We have demonstrated the effectiveness of articulatory style vectors in supplementary materials.

### 3.4 Parameter Prediction

As shown in Fig. 3, the TTS synthesis module of *ArtSpeech* has three predictors, including the duration predictor, the articulatory predictor, and the mel spectrogram predictor. All these predictors have a similar encoder-decoder structure and take phonemes extracted from the text as input. Style vectors obtained from articulatory features will respectively modulate corresponding predictors to achieve the synthesis of specific timbres and rhythms.

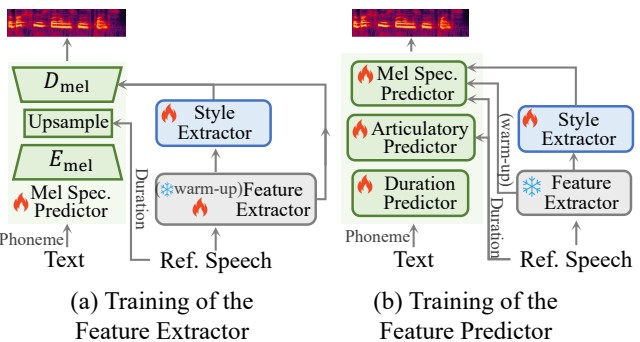

Figure 3: (a) TV Extractor, estimates the 10-d TVs from the mel spectrogram. (b) Architecture of our *ArtSpeech*, including style extraction (Sec. 3.3) and parameter prediction (Sec. 3.4).

(a) Training of the Feature Extractor

(b) Training of the Feature Predictor

Figure 4: The training strategy of *ArtSpeech*, Figure (a) shows the training stage of the feature extractor, where the main difference between the warm-up step and the subsequent step is whether the parameters of the articulatory feature extractor are fixed or not. Figure (b) shows the training stage of the feature predictor, where the main difference between the warm-up step and the subsequent step is whether the articulatory features are extracted from the feature extractor or predicted from the text.

**Duration Predictor**, uses a two-layer feed-forward attention layer as an encoder to extract phoneme features, a three-layer AdaIN [23] block as a decoder, and a Bi-LSTM layer followed by a linear layer. The predictor takes the TVs as input and outputs duration style vector which is fed into the AdaIN block to help with speaking style transfer in terms of speed.

**Articulatory Predictor**, performs the phoneme-level articulatory parameter prediction, *i.e.* energy, $F_0$, and TVs, sharing the same feed-forward attention-based encoder. The encoded results are aligned by the predicted duration and then fed into different decoders along with corresponding style vectors. Each articulatory style vector is individually applied to its corresponding predictor to ensure that the estimated articulatory parameters align with the target style. The decoder architecture is similar to that of the duration predictor. We will further illustrate the effectiveness of this

predictor in Sec. 5.2 and Sec. 6 – enhancing the prosody similarity of synthesized speeches to the target speech and improving the controllability of the synthesized results.

**Mel Spectrogram Predictor**, adopts a similar architecture to the other two predictors. The encoder features a four-layer feed-forward attention. Along with two kinds of style vectors, the aligned results are fed into the six-layer AdaIN blocks. Specifically, in the first three layers, the output of the previous layer is concatenated with articulatory style vectors and mel spectrogram style vector as the input for the subsequent layer. The last three layers only utilize the output of the previous layer as input and employ mel spectrogram style vectors for style transfer. Finally, a linear layer is used to synthesize the final spectrogram.

## 3.5 Training

To enhance the accuracy of articulatory feature extraction and prediction, and to ensure the physical significance of these features within the model, we divide the training process into two main steps: training of the articulatory feature extractor and training of the articulatory feature predictor.

**Training of the Feature Extractor** As shown in Fig 4(a), during this stage, the articulatory features extracted from the feature extractors are not only used for extracting style vectors but also serve as inputs to the mel spectrogram predictor. The ground truth duration is extracted by [35].

Since the feature extractors are pre-trained, we employ a certain number of warm-up steps during which the parameters of the feature extractors are fixed, focusing on training the mel predictor first. Three loss functions are taken into account to facilitate the training of the mel spectrogram predictor: reconstruction loss $\mathcal{L}_{recon}$; adversarial loss $\mathcal{L}_{adv}$ for training an additional discriminator; and feature mapping loss $\mathcal{L}_{fm}$ for improving the speech quality [28].The loss function is defined as follows:

$$\mathcal{L}_{mel} = \lambda_{L1}\mathcal{L}_{recon} + \lambda_{adv}\mathcal{L}_{adv} + \lambda_{fm}\mathcal{L}_{fm},$$
$$\mathcal{L}_{recon} = \|x - \hat{x}\|_1,$$
$$\mathcal{L}_{adv} = \log D(x) + \log(1 - D(\hat{x})),$$

$$\mathcal{L}_{\text{fm}} = \sum_{l=1}^{T} \frac{1}{N_l} \left\| D^l(x) - D^l(\hat{x}) \right\|_1,$$

where $x$ and $\hat{x}$ represent the original and reconstructed mel spectrogram, respectively. $D^l$ represents the feature map of the $l$-th layer of the discriminator $D$ with $N_l$ number of features, $T$ is the total number of layers in the discriminator.

After the warm-up steps, we release the fixed parameters of the feature extractors. To ensure the articulatory features retain their physical significance during training, we apply regularization losses $\mathcal{L}_{\text{reg}}$ to constrain the training process of both $F_0$ and TV feature extractors. We conduct ablation studies to demonstrate the effectiveness of these losses in supplementary materials. The loss function in this step is defined as follows:

$$\mathcal{L}_{\text{Ext}} = \mathcal{L}_{\text{mel}} + \lambda_{\text{reg}} \mathcal{L}_{\text{reg}} + \lambda_{\text{s2s}} \mathcal{L}_{\text{s2s}} + \lambda_{\text{mono}} \mathcal{L}_{\text{mono}},$$

where

$$\mathcal{L}_{\text{reg}} = \left\| f_0 - f_0' \right\|_1 + \left\| v - v' \right\|_1 .$$

$f_0$ and $v$ represents the extracted features extracted by the initial pre-trained model; $f_0'$ and $v'$ represents the new predicted results in this step. We utilize the model and the corresponding losses proposed by Li *et al.* [35] to extract speech duration. $\mathcal{L}_{\text{s2s}}$ is the sequence-to-sequence loss to ensure correct attention alignment; and $\mathcal{L}_{\text{mono}}$ aims to force the soft attention alignment to be close to its monotonic version. Please refer to [35] for more details.

**Training of the Feature Predictor**

As shown in Fig. 4(b), during this stage, we fix the parameters of the trained articulatory feature extractors and involve the feature predictors and duration predictor in the model training. This stage also includes a certain number of warm-up steps, during which we continue to utilize the features inputs extracted from reference mel spectrograms for the mel spectrogram predictor, rather than using the predicted results from text by these feature predictors. By independently training the feature predictors using L1 loss, we can accelerate the convergence speed of the parameters. The loss function for this training step is defined as follows:

$$\mathcal{L}_{\text{Pre}} = \mathcal{L}_{\text{mel}} + \lambda_{\text{p}} \mathcal{L}_{\text{p}},$$
$$\mathcal{L}_{\text{p}} = \sum_{\theta \in \Theta} \lambda_\theta \left\| \theta_{gt} - \theta_{pred} \right\|_1.$$

$\Theta = (\theta_d, \theta_e, \theta_{f_0}, \theta_v)$ denotes the set of features including duration, energy, $F_0$, and TVs. Specifically, $\theta_{gt}$ represents the results extracted from the reference speech $r$ and $\theta_{pred}$ represents the results predicted by predictors.

After the warm-up steps, We employ these parameter predictors to predict articulatory parameters from text, which are then inputted into the mel spectrogram predictor. Conversely, features extracted from the reference mel spectrogram are solely utilized for extracting the style vectors. We fine-tune the rest of the model to improve blocks' co-adaptation.

# 4 EXPERIMENTS

## 4.1 Dataset

We conducted experiments on two public datasets:

- LibriTTS dataset [65], used to evaluate the zero-shot adaptive TTS synthesis results of our *ArtSpeech*. We screen the *train-clean-360* and *train-clean-100* subsets of LibriTTS for clips longer than 1 second and shorter than 20 seconds. In total, approximately 245 hours of speeches recorded by 1,151 speakers are used in the training process. We use the same split strategy of data as StyleTTS [35], *i.e.* 98% for training, 1% for validation, and 1% for testing. We utilize the *test-clean* subset of LibriTTS for the evaluation of *ArtSpeech*.

- LJSpeech dataset [25], used to evaluate the synthesized speeches in terms of audio quality and similarity. It consists of approximately 24 hours, totaling 13,100 clips recorded by a single speaker. We split the dataset into 12,500 samples for training, 100 for validation, and 500 for testing. Moreover, all clips are upsampled to a 24 kHz sampling rate.

All audio clips are transformed into 80-d mel spectrograms using the Fast Fourier Transform Algorithm (FFT), with 2048 FFT size, 1200 window size, and 300 hop size. We use the International Phonetic Alphabet (IPA) to label phonemes and phonemizer package [2] to convert text into IPA sequences.

## 4.2 Implementation

All modules are trained using AdamW optimizer [40] with beta schedule of $\beta_1 = 0$, $\beta_2 = 0.99$, and $\epsilon = 10^{-9}$. We employ the cosine annealing schedule for learning rate decay with an initial learning rate of $2 \times 10^{-4}$ and a minimum value of $4 \times 10^{-5}$.

For the training process of the LJspeech dataset, the training of both the feature extractor and feature predictor was executed for 120 epochs, with the warm-up step both accounting for 20 epochs. while for the LibriTTS dataset, the training of the feature extractor stops after 40 epochs. We configure loss weights as: $\lambda_{L1} = 5$, $\lambda_{adv} = 1$, $\lambda_{fm} = 0.1$, $\lambda_{reg} = 1$, $\lambda_{s2s} = 1$, $\lambda_{mono} = 1$, $\lambda_p = 1$. Different articulatory features have varying scales. After completing the pre-training of the articulatory feature extraction model in Sec. 3.3, we pre-extract the articulatory features from the dataset and conduct statistical analysis on the overall mean and variance. During the calculation of loss in subsequent training phases, we normalize the estimated results. This ensures uniformity in the loss scale among various articulatory features.

## 4.3 Evaluation

We recruited 570 native speakers via Amazon Mechanical Turk (MTurk) to use a 5-point Likert scale to rate all the synthesized speeches mentioned below, with 1 meaning bad performance and 5 meaning the opposite.

- To evaluate how well our *ArtSpeech* performs in zero-shot speaking style transfer for Out-of-Domain (OOD) speakers, we use the official implementations and pre-trained models or released demos of YourTTS [5], StyleTTS [35] and StyleTTS2 [36] as comparison approaches. We also use an open-source implementation of Microsoft's VALL-E X [69] zero-shot TTS model. Specifically, we leverage the subset of the LibriTTS dataset (*LibriTTS-test-clean*) and randomly select two clips as reference speech of each speaker to conduct TTS synthesis. We asked participants to rate the Mean Opinion Score of Overall Similarity (MOS-O) between the reference speeches and the synthesized results.

**Table 1: Evaluation results of overall similarity ratings (MOS-O) on the LibriTTS dataset with 95% CI.**

| Model | MOS-O |
|---|---|
| Ground Truth | 4.12±0.06 |
| YourTTS + HiFi-GAN | 3.76±0.09 |
| VALL-E X | 3.68±0.08 |
| StyleTTS + HiFi-GAN | 3.89±0.07 |
| StyleTTS 2 | 4.01±0.06 |
| ArtSpeech + HiFi-GAN | **4.08±0.07** |

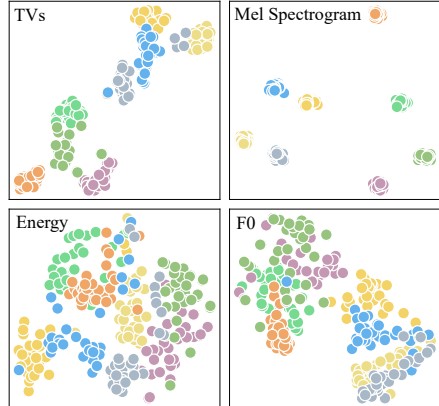

**Figure 5: Style features of randomly selected 30 speeches from 8 speakers in the LibriTTS dataset, dimensionality reduced into 2D planes for better visualization.**

- To further evaluate the similarity of the synthesized speech to the target speaker's voice tone and prosody (MOS-S), we randomly select 50 texts to conduct TTS synthesis and compare the performance of *ArtSpeech*, FastSpeech2 [47], VITS [28], StyleTTS [35], and StyleTTS2 [36], all trained on the LJSpeech dataset. We also asked participants to rate the audio quality (MOS-Q).

Note that all speeches are randomly presented to participants to avoid bias. We also conduct an ablation study to evaluate the effectiveness of our vocal track variables (TVs), model design, training strategy, etc. Please refer to supplementary materials for detailed procedures and the corresponding questionnaires.

For quantitative evaluation, we generated speech that is consistent with the reference speech content and compared the correlation coefficients (Corr) and Mean Absolute Error (MAE) metrics of $F_0$ and energy between the ground truth and speeches synthesized by different approaches. This metric serves to elucidate the prosody similarity between different speeches.

## 5 RESULTS

### 5.1 Zero-Shot Speech Synthesis

We calculate the Mean Opinion Score (MOS) of participants' ratings for the overall similarity (MOS-O) of the speeches synthesized

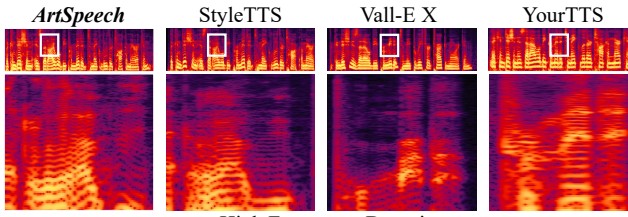

High Frequency Domain

**Figure 6: Zoom in on the high-frequency domain of synthesized speeches with the same text ("that"). Our result shows clearer formant frequencies (the bright yellow bands) and less absent information.**

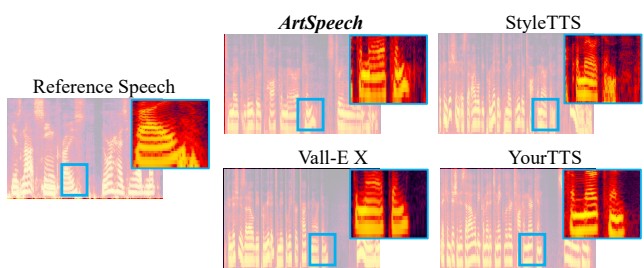

**Figure 7: Zoom in on the part between two sentences in speeches. Our result shows a similar intonation to the reference speech, *i.e.* showing a rising pitch followed by a short falling pitch.**

**Table 2: Evaluation results on the LJSpeech dataset of voice tone and prosody similarity (MOS-S) and speech quality (MOS-Q), completing with 95% confidence intervals (CI).**

| Model | MOS-Q | MOS-S |
|---|---|---|
| Ground Truth | 3.98±0.07 | - |
| FastSpeech 2 + HiFi-GAN | 3.44±0.10 | 3.58±0.08 |
| VITS | 3.73±0.08 | 3.97±0.06 |
| StyleTTS + HiFi-GAN | 3.87±0.07 | 3.89±0.07 |
| StyleTTS 2 | 3.91±0.07 | 4.02±0.07 |
| ArtSpeech + HiFi-GAN | **3.96±0.06** | **4.06±0.06** |

by different models. As shown in Table 1, our *ArtSpeech* achieves better results, *i.e.* 4.12 ± 0.05, following the ratings of ground truth (4.18 ± 0.05) and clearly outperforming other models. Additionally, we plot our four style vectors of 8 randomly selected speakers, projected into 2D planes (Fig. 5). The results show that our model excels well in representing the speaking styles of different speakers, particularly for TVs style and mel spectrogram style. The relatively higher overlapping of energy style and $F_0$ style could be explained by the three steps of speech production – different speakers may have similar expelled air and vocal cord vibration frequencies; yet the resonant cavities formed by multiple articulatory organs are relatively unique. Overall, the results demonstrate *ArtSpeech*'s better performance under both multi-speaker and zero-shot settings.

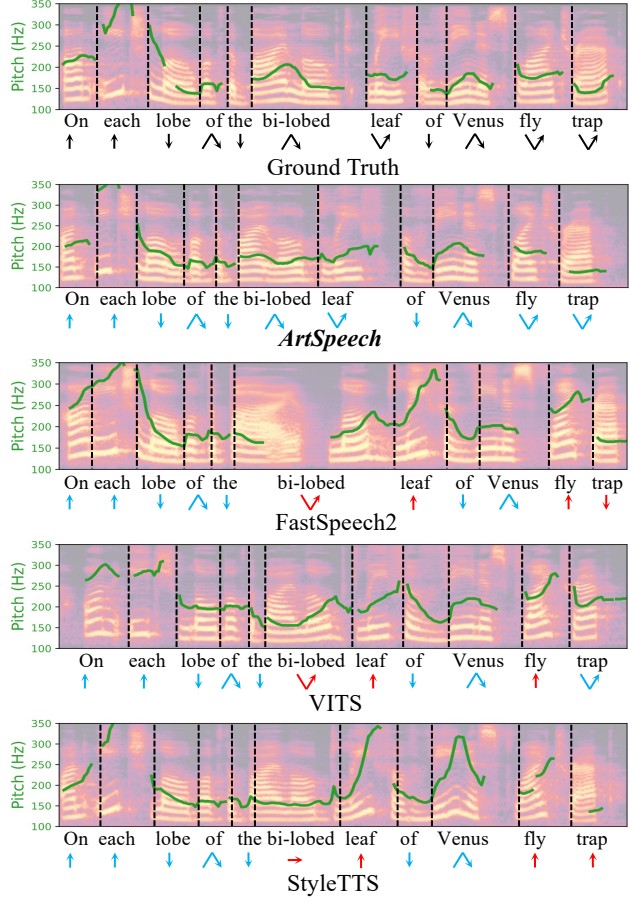

**Figure 8: Mel spectrograms different speeches with $F_0$ curves (green lines) and intonations of the corresponding words. *ArtSpeech*'s results are closer to the ground truth than others. (The blue arrows represent intonations consistent with the ground truth; the red arrows mean the opposite.)**

We further illustrate the effectiveness of our *ArtSpeech*. **First**, as shown in Fig. 6, our synthesized speech exhibits clearer formant frequencies (the bright yellow bands), especially in the high-frequency domain. Our result also has less absent information compared to that of Vall-E X [69]. This highlights *ArtSpeech*'s capacity to understand and mimic the pronunciation process, converting the input text to human-like speech. **Second**, our synthesized speech has an $F_0$ curve resembling that of the reference speech (Fig. 7). This supports that *ArtSpeech* could capture the speaking patterns of the target speaker, *e.g.*, always showing a rising pitch followed by a short falling pitch during two sentences in the speech. On the contrary, other speeches have less similarity to the reference speech, *i.e.* no subtle changes in such circumstances.

## 5.2 Speech Quality and Similarity

As shown in Table 2, *ArtSpeech* performs better than other models in both speech quality and similarity (with 95% confidence intervals (CI)), *i.e.* $3.92 \pm 0.07$ (MOS-Q) and $4.05 \pm 0.08$ (MOS-S), approaching

**Table 3: The correlation coefficients (Corr) and Mean Absolute Error (MAE) metrics of F0 and energy between the reference speech and synthesized speech generated by each model.**

| Method | $F_0$ | | Energy | |
|---|---|---|---|---|
| | MAE ↓ | Corr ↑ | MAE ↓ | Corr ↑ |
| *LibriTTS* | | | | |
| YourTTS + HiFi-GAN | 19.31 | 0.85 | 1.18 | 0.81 |
| VALL-E X | 28.36 | 0.77 | 0.81 | 0.87 |
| StyleTTS + HiFi-GAN | 17.98 | 0.86 | 0.61 | 0.93 |
| StyleTTS 2 | 16.26 | 0.87 | 0.58 | 0.93 |
| ArtSpeech + HiFi-GAN | **15.89** | **0.89** | **0.56** | **0.95** |
| *LJSpeech* | | | | |
| FastSpeech2 + HiFi-GAN | 26.05 | 0.85 | 0.75 | 0.91 |
| VITS | 28.35 | 0.84 | 0.62 | 0.94 |
| StyleTTS + HiFi-GAN | 26.99 | 0.83 | 0.64 | 0.94 |
| StyleTTS 2 | 27.43 | 0.84 | 0.68 | 0.89 |
| ArtSpeech + HiFi-GAN | **23.83** | **0.87** | **0.61** | **0.95** |

the ratings of ground truth. The results support that our generated speeches have reduced noise and express a voice tone and prosody similar to that of the target speaker. In Table 3, we demonstrate the results of correlation coefficients and MAE metrics of $F_0$ and energy respectively. For both test sets of the multi-speaker dataset (LibriTTS) and single-speaker dataset (LJSpeech), our *ArtSpeech* synthesized results have a relatively similar prosody expression to that of the ground truth.

Fig. 8 further visualizes some generated results with fundamental frequency ($F_0$) curves and intonations. All speeches says *"On each lobe of the bi-lobed leaf of Venus flytrap."* We observe that *ArtSpeech* generated results closely consistent with the ground truth; the $F_0$ curves are similar, both of which are smooth and coherent; the intonation changes are consistent.

## 5.3 Ablation Study

We conducted ablation experiments to validate the effectiveness of our *ArtSpeech* in terms of the selected vocal track variables (TVs), model design, training strategy, etc. Experiments are carried out on the single-speaker dataset, LJSpeech. We leverage the Comparison Mean Opinion Score (CMOS) to organize participants' ratings. The results are shown in Table 4. All settings lead to a reduction in ratings. For example, we remove the TV-related module, and the ratings drop by 0.20; we simplify the style extractor and concatenate articulatory features and mel spectrogram, leading to a 0.11 reduction; we delete the additional encoders in the articulatory and duration predictors and CMOS drops by 0.32 and 0.19, respectively; To validate the effectiveness of our training strategy (Sec. 3.5), we first remove the $L_{reg}$ loss from the first training step, *i.e.* without any explicit constraints during $F_0$ and TVs feature extractor training. This leads to a substantial decrease of 0.60. In supplementary materials, we plot the results of articulatory parameter predicted from *ArtSpeech* and w/o $L_{reg}$ settings. We observe that there is a significant amount of irregular noise in the latter case. Additionally,

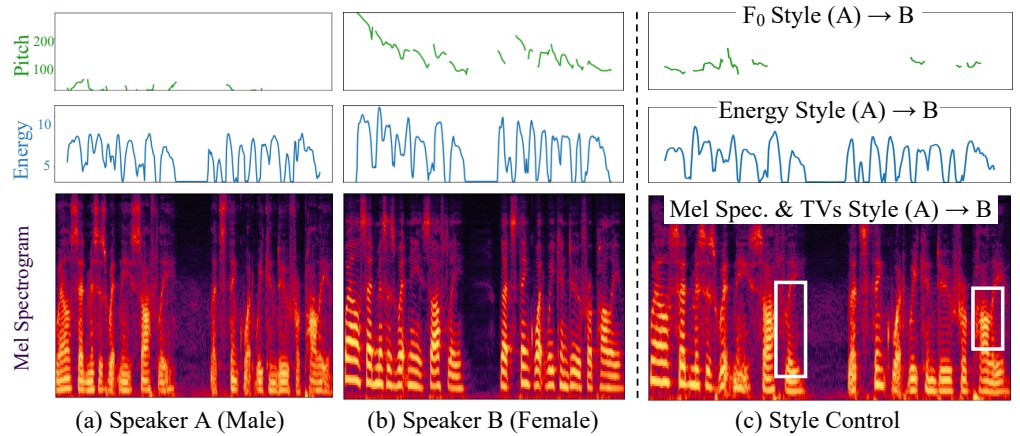

**Figure 9: Visualizations of synthesized speeches with different style controls, *i.e.* we incorporate style vectors (Speaker A) of $F_0$, energy, and mel spectrogram and TVs into that of Speaker B, respectively. The results show a shift in Speaker B's speaking style towards that of Speaker A, *e.g.* lower $F_0$ and energy, as well as changed intonations and reduced information in the high-frequency domain.**

**Table 4: Ablation study results. All settings lead to rating drops.**

| Setting | CMOS |
| --- | --- |
| ArtSpeech | 0 |
| w/o TVs | -0.20 |
| w/o multi-style | -0.11 |
| w/o articulatory encoder | -0.32 |
| w/o duration encoder | -0.19 |
| w/o $L_{reg}$ loss | -0.60 |
| w/o warm-up step of the second stage | -0.14 |

we eliminated the warm-up step of the second training step and fine-tuned the entire model directly, leading to a 0.14 drop.

## 6 DISCUSSION

In this section, we attempt to demonstrate the speaking style controllability of the proposed *ArtSpeech*. As our multi-dimension style extractor decouples the speaking style of the target speaker (Sec. 3.3), *ArtSpeech* could employ independent articulatory style vectors to facilitate flexible style combination and degree adjustment. We encourage readers to view the demo page to hear the corresponding synthesized speeches.

As shown in Fig. 9, we visualize two very different speakers' speeches: Speaker A (Fig. 9(a)), male, has a relatively deeper voice with low fundamental frequency ($< 150Hz$) and energy ($< 8$), and less high-frequency domain information in mel spectrogram; Speaker B (Fig. 9(b)), female, has a brighter voice with high fundamental frequency ($> 150Hz$) and energy ($> 8$), and more formant frequencies in high-frequency domain in mel spectrogram.

In Fig. 9(c), we demonstrate the results of incorporating different style vectors of the male (Speaker A) into the speech of the female (Speaker B). To specifically drop the female's pitch, we add the

male's $F_0$ style vector during the synthesis and achieve a dropped $F_0$ curve, while other styles remain unchanged. Similarly, we add different style vectors into the female's speech and the synthesized results sound more like the male's voice tone and prosody. The results support that our articulatory style vectors could help with shaping different synthesized speeches. In supplementary materials and the demo page, we demonstrate the synthesized speeches and also provide a visualized plot of continuous style degree adjustment, *i.e.* incorporating varying degrees of the male's style vector into the female's speech.

## 7 CONCLUSIONS

We introduce *ArtSpeech*, a text-to-speech (TTS) system, building upon the articulatory system's physical significance, that enables zero-shot style transfer of custom voices outside of its domain. Compared to other deep leaning-based models, *ArtSpeech* has merits in: i) systemic interpretability by bringing the intuitive articulatory representations into TTS frameworks; ii) break the bottleneck of limited articulatory data, facilitating articulatory feature extraction from the input text; iii) experiments confirm the effectiveness and enhanced speaking style similarity between synthesized speeches to target speakers' speeches. For future work, We believe there is room for improvement in modeling speakers' unique pronunciation styles while eliminating extraneous environmental noise interference.

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
