# OpenReview forum: "ArtSpeech: Adaptive Text-to-Speech Synthesis with Articulatory Representations"
_acmmm.org/ACMMM/2024/Conference — MM2024 Oral_

### Official Review · Reviewer_9JGG · 2024-05-22

**Rating:** 4
**Confidence:** 4

**Summary:**

The paper proposed an articulatory representation-based text-to-speech (TTS) model, ArtSpeech for expressive and adpative TTS. This model leverages the explicit physical significance of articulation movement and designs a multi-dimensional style mapping network to extract speaking styles. Experimental results demonstrate that compared to other open-source zero-shot TTS systems, ArtSpeech enhances synthesis quality and greatly boosts the similarity between the generated results and the target speaker’s voice and prosody.

**Strengths:**

1) It leverages articulatory representations to perform expressive and adaptive TTS. Such a new perspective make the information related to formants like timbre interpretable.

2) The authors conducted comprehensive experiments and analyses, and describes the experimental details well. Detailed supplementary materials are also provided for the experiments. The sujective results are convicing with 95% confidence intervals. The objective experiments are very intuitive and detailed. The model is proved to perform well under zero-shot speech synthesis, speaker adaptation, style control tasks. The audio demo also supports their conclusion well.

**Limitations:**

1) The framework of the proposed model is common and does not describe more improvements for articulatory data. Actually, it looks like giving another ground truth data, and the model only performs a fitting task from the extracted features. This is where I feel that there is insufficient innovation.
2)  The leverage of EMA is one of my considerations. And I'm confused about using  7.9 hours of 44.1 kHz speeches and 100 Hz EMA data recorded by 8 participants to pre-train TV extractor is whether enough even under zero-shot condition. Hence, the generalizability of this module is what I also consider about.
3) In Section 3, the logical coherence is not strong enough. It is more like describing how to do in each module without saying why it should be done. Also readers want to understand the whole picture more intuitively in main text or supplementary materials, rather than referring to the pre-knowledge provided by other documents in subsection 3.5.
4) The modeling of f0 and energy is significant in prosody modeling.  The regularization loss \mathcal{L}_{reg} contributes a lot into performance due to the ablation study results. I would like to know how the performance will degrade without (f_0-f'_0) but preserving (v-v').
5) In Introduction Section, it lacks a summary of article contributions.
6) The subtitle in Section 6 is not clear. If you want to demostrate the model performance in specific task, just name it after this.

**Suitability:**

2

---

### Official Review · Reviewer_vfqw · 2024-05-23

**Rating:** 5
**Confidence:** 3

**Summary:**

This paper investigates extraction of style information from articulatory features for generating speech with specific styles from reference speech utterances. An articulatory feature extractor is first trained on the HPRC dataset. The extractor is then used to extract articulator features, including tongue tip, tongue blade etc, which are fed to the style extractor, together with extracted F0, energy, to provide style information for the synthesis model.

**Strengths:**

The proposed framework is novel. Considering articulatory features that are related to pronunciation is promising for speech synthesis. Though the conclusion is in expectation as previous works also have tried to integrate articulatory features in speech synthesis, it is still good to demonstrate that articulatory features can enhance expressive speech synthesis.

The key idea of this paper is training an articulatory feature predictor from an external dataset. This predictor is then used as an initialized point in the style vector extraction and further finetuned with the target domain dataset, i.e., LibriTTS and LJSpeech.

To alleviate the affect of dataset shifting, the current proposal adapts the TV extractor towards mel-spectrogram prediction with a regularization term that constrains the adapted prediction to be close to the original prediction. Experimental results in Table 4 demonstrate the effectiveness of the regularization term L_reg.

Experimental results on extracting style information from reference speech as articulatory features demonstrate that the proposed framework is effective.

**Limitations:**

line 257 se=𝑆𝐸(𝑒, 𝑓0, 𝑣, 𝑟 ), should be sv=SE(.)?

How accurate is the TV extractor, i.e., on the HPRC dataset? How does the accuracy affect the speech synthesis performance?

The TV extractor is trained on external HPRC dataset and applied to the target dataset of LibriTTS or LJspeech, it may cause significant performance degradation because of the different characteristics of datasets. Considering this domain shifting is important. The current version only support this by the ablation study in Table 4. Since the target datasets don't contain articulatory features, it would be helpful to include more analysis on TV extraction accuracy on the target dataset in some way, e.g., examining the relationship between articulatory features and the prosodic features of F0, energy, on the external dataset and the target datasets.

The HPRC dataset only contains 8 participants, is it sufficient for build the speaker-independent TV extractor? How is the speaker normalization conducted to ensure correct TV extraction?

Introduction section mentions the motivation of data efficiency. Does the proposed articulatory feature-based approach have better data efficiency than previous approaches?

Controllability is another motivation of this paper. Is it possible to control synthetic prosody/style through the articulatory features? It would be helpful to show the controlling not only by the prosodic features, but also via the integrated articulatory features.

**Suitability:**

2

---

### Official Review · Reviewer_vU2v · 2024-05-24

**Rating:** 6
**Confidence:** 2

**Summary:**

The paper proposed a parametric text to speech model with an intrinsic interpretable module that learns speaker-wise parameters for energy, f0, and the vocal tract. In a first stage the parameters are trained on a reference voice, then with a second network the style is learned and the speech is synthesized.

**Strengths:**

A major strength and novelty for the paper is the fact that they propose an interpretable method with parameters that can be controlled. The paper presents experimental proof for the validity of the method and an ablation study. I found the text to be clear although the authors give plenty of details for the method. The demo presented in the supplementary materials are convincing.

**Limitations:**

Issues:
- Are F0, duration, VT learned for each speech unit (phoneme, vowels) for a given speaker?
- The method estimates three sets of parameters (duration, articulatory, mel) using three different networks. The choice of the layers and modules should be better advocated for and explained.
- no open source implementation given that so many existing methods are open-source already. this can limit the impact of this paper :(

Ideas:
- maybe use this clean version of the librispeech since the original is not exactly clean and may lead to errors in parameter estimation: https://www.openslr.org/141/
- a comparison with an opensource version of soundstorm would be desirable or a re-training of some existing methods with the same data, since the experimental conditions are different in each of the papers


Minor issues:
- why is "sound and music computing" listed as a domain?
- when describing limitations of previous methods in the intro, it is good to give some references to the specific methods that present those limitations
- what is an articulatory representation and how is the different from a physical model such as source-filter model?
- sv and se are not good math notations for vectors; maybe decide for a single letter
- in 3.2 it seems that you are describing exactly the source-filter model with a more complex definition for the filter (TT, TB etc.)
- this is not very clear. what data is used for testing? ".e. 98% for training, 1% for validation, and 1% for testing. We utilize the test-clean subset of LibriTTS for the evaluation of ArtSpeech"
- in Figure 8 the actual pitch is not plotted to match the scale of the spectrogram

**Suitability:**

3

---

### Official Review · Reviewer_v5uN · 2024-05-24

**Rating:** 3
**Confidence:** 4

**Summary:**

This paper proposes a text-to-speech model named ArtSpeech. It incorporates articulatory and prosodic features as conditions for style speech synthesis. The model uses a multi-stage training strategy to incorporate the articulatory feature extractor and the main text-to-speech modeling. Subjective evaluations and qualitative analysis are conducted to demonstrate the effectiveness of the proposed method.

**Strengths:**

- The paper introduces good intuitions about using articulatory features to facilitate style TTS. This could potentially complement style information derived from commonly used features such as F0, energy, and Mel-spectrogram. The proposed method uniquely incorporates a 10-dimensional articulatory feature that represents the distances and constriction degrees between different points in the vocal tract.
- The overall presentation is clear and easy to comprehend. Each component is well-explained with appropriate technical details.

**Limitations:**

### **Regarding evaluation**:
- Lack of objective metrics. While the authors conducted extensive subjective evaluations regarding the overall similarity, quality and prosody similarity, it lacks objective evaluation over speaker similarity, word error rates, etc. These objective metrics could be crucial for the community to make comparisons.
- Lack of analysis regarding the incorporation of the vocal tract features. The ablation study shows the removing of the vocal tract features incur a degradation of overall MOS. While it is a novel feature to utilize in TTS, the paper does not conduct specific study to demonstrate what aspect of TTS is benefited from the introduction of these features.
- Some qualitative analysis is ambiguous. Figure 5 visualizes different style vectors and shows that they cluster by speaker ID. However, this doesn't necessarily imply good representation as these features are speaker-dependent. Figure 7 aims to show the similarity between the synthesized speech and the generated speech in terms of F0, but it can be hard to find the local temporal correspondence between the reference and the synthesized speech when the reference speech is condensed into a single vector. This analysis and comparison can sometimes be misleading.
- Regarding Table 3, are the reference speech the same as the target speech? If not please specify how to compute the MAE and Corr for F0 and energy.
### **Regarding modeling**:
- The paper lacks novelty in modeling. Although ArtSpeech incorporates new features as input, the modeling structures are very common in TTS.

### **Regarding paper writing**:
- Typos: 1) "JCD" in Figure 3 (a) should be "JDC"; 2) The two MOS scores in section 5.1 are inconsistent with those in Table 1; 3) In section 5.3, "vocal track" should be "vocal tract".

**Suitability:**

2

---

### Meta-Review · Area_Chair_Kryo · 2024-07-01

**Recommendation:** Accept (Oral)
**Confidence:** 4

**Metareview:**

This paper presents an approach to controllable TTS using modeling of style information by articulatory and phonetic features for generating speech with specific styles from reference speech utterances. 10-dimensional articulatory feature extractors are being trained on the HPRC dataset, which are then used to extract tongue tip, tongue blade position etc (in addition to energy, f0), and give an interpretable speaker-specific background model, which can then be adapted to style. Authors present perceptive as well as objective metrics and demonstrate that the proposed approach enables good quality synthesis while being interpretable and controllable. There is little work on TTS with articulatory features, so the proposed work has novelty and does fit the wider scope of ACM MM.

Reasons to accept:
- The proposed framework is novel and a good addition to the body of work on expressive speech synthesis.
- The work leverages the pre-training then fine-tuning paradigm and can thus be compared to many other works e.g. on LibriTTS or LJSpeech data.
- Ablations show that extraction of style information is effective.
- The regularization term L_reg that constrains the strength of the adaptation is shown to be effective in detailed ablations (Table 4)

Reasons to reject:
- Reviewers had several technical questions (e.g. accuracy of TV extractor, domain generalization, speaker normalization), but these were addressed largely in the rebuttal; it is assumed the paper will be updated accordingly.
- Some questions remain on the size of the data sets (8 speakers?) Does the model indeed have better data efficiency than earlier approaches?
- Since controllability is the core aspect of the paper, it would be helpful to show the controlling not only by the prosodic features, but also via the integrated articulatory features.